# Inferring incompetence from employment status: An audit-like experiment

**Celestin Okoroji** [ID] [1,2], **Ilka H. Gleibs** [ID] [1]*, **Simon Howard** [3]

**1** Department of Psychological and Behavioural Science, London School of Economics, London, United Kingdom, **2** BlackThrive Global, London, United Kingdom, **3** Department of Psychology, University of Miami, Coral Gables, Florida, United States of America

* i.h.gleibs@lse.ac.uk

**Data Availability Statement:** All data, materials and code for the studies within this paper are available from the Open Science Framework (https://osf.io/rf7hw/).

## Abstract

Audit studies demonstrate that unemployed people are less likely to receive a callback when they apply for a job than employed candidates, the reason for this is unclear. Across two experiments ($N = 461$), we examine whether the perceived competence of unemployed candidates accounts for this disparity. In both studies, participants assessed one of two equivalent curriculum vitae's, differing only on the current employment status. We find that unemployed applicants are less likely to be offered an interview or hired. The relationship between the employment status of the applicant and these employment-related outcomes is mediated by the perceived competence of the applicant. We conducted a mini meta-analysis, finding that the effect size for the difference in employment outcomes was $d = .274$ and $d = .307$ respectively, while the estimated indirect effect was -.151[-.241, -.062]. These results offer a mechanism for the differential outcomes of job candidates by employment status.

## Introduction

Unemployment can lead to relative, and in some cases, absolute poverty effecting housing, food consumption and leisure activities [1]. Consequently, either directly or indirectly, unemployment has serious psychological consequences in terms of well-being, self-esteem and cognitive performance [2–4]. Relatedly, unemployed persons face high levels of stigmatisation [5, 6].

Previous research has shown that unemployed people themselves are aware of the stereotypes that others hold about them and show low levels of identification with unemployment [7]. However, stigmatisation does not only affect the target of stigmatisation but also those perceiving the target. This can lead to unemployed people being assessed less favourably in the job market compared with similarly qualified employed people [8–10]. What is less well understood, are the mechanisms that lead to this discrimination in the job market. From earlier research, we know that unemployed people have been shown to believe that others perceive them as less competent than they see themselves [7]. If the perception unemployed people have about others' views of their competence is accurate [11], employers may be more likely to

**Funding:** The author(s) received no specific funding for this work.

**Competing interests:** The authors have declared that no competing interests exist.

see unemployed people as less competent than equally qualified employed candidates. Hence, perceived competence may differ by employment status resulting in differential job market outcomes. In the current research, we examine whether perceptions of competence mediate the relationship between employment status and employment-related outcomes.

## Audit studies

There is extant literature that has examined the effects of unemployment on job market outcomes. In general, these studies use a broadly similar methodology [12], to test whether unemployed (vs. employed) individuals face bias due to their employment status. Typically, resumes/curriculum vitae's (henceforth CVs) are created which are identical except for the employment status of the applicant and sent out to real job vacancies. Callback rates are then recorded, and bias is demonstrated when there is a significant difference in callbacks by employment status. However, although numerous audit studies have documented that unemployment reduces the likelihood of a callback and provide very strong evidence for discriminatory practices [8, 13–16], what is less clear from these studies is the underlying psychological processes contributing to this bias. Although many theories have been put forth as to why unemployed people receive fewer callbacks, due to ethical concerns [16, 17], audit studies are limited in the extent they can answer questions about the mechanisms which contribute to bias in hiring and selection.

As such, the audit method does not allow direct assessment of the psychological processes (e.g., stereotypes) that influence recruiters' decisions. Additionally, with more hiring and selection processes moving online, another limitation of audit studies is that many organisations no longer accept CVs and this varies systematically by industry [18]. To address the limitations of the audit method, the present study uses an online 'audit-like' experiment, which mimics audit methodology and allows us an avenue to investigate mechanisms underlying biased outcomes. One such mechanism may be the perceived competence of the applicant.

## Perceived competence as a mediator between employment status and employment bias

The stereotype content model [19–22] suggests that three basic dimensions underpin group stereotypes. These are competence, warmth [20] and morality [22]. In the context of hiring decisions, in which organisations seek to employ the most productive staff, an employer's perception of candidate competence is likely to influence their decisions about whom to interview and ultimately hire. Thus, the competence dimension of the stereotype content model offers a plausible social-psychological mediator of the poor employment outcomes that have been documented in previous research [10, 23]. Specifically, we hypothesise that unemployed people are seen as less competent than employed people, which contributes to the finding that unemployed people receive fewer callbacks. To our knowledge, no studies to date have directly examined the perceived stereotype content of job applicants and its relation to progression through the application process.

However, this hypothesis cannot be assessed using the audit method. Thus, to further our understanding of the mechanisms which contribute to differential outcomes for unemployed applicants, we argue for online experimentation to understand the relationship between unemployment and job market outcomes.

## The present study

In the present study, we compare an unemployed candidate to a currently employed candidate with the same experience and qualifications to assess the effects of unemployment on various

employment-related outcomes. Specifically, we examine the likelihood that the candidate will be interviewed and hired. Importantly, we include stereotype content measures [22] which allow us to examine differences in morality, warmth and competence and test if employment outcomes are mediated by the stereotype content model dimensions, in particular competence.

All data, materials and code for the studies within this paper are available from OSF (https://osf.io/rf7hw/).

**Hypotheses.** Based on the extant literature we hypothesise that:

$H_1$—The unemployed candidate will be less likely to be interviewed than an equivalent employed candidate

$H_2$—The unemployed candidate will be less likely to be offered employment than an equivalent employed candidate

$H_3$—The relationship between employment status and employment outcomes will be mediated by perceived competence.

## Study 1

### Method

**Participants and design.** One hundred and eighty-seven participants completed an online experiment on prolific academic [www.prolific.ac] and were paid £7.50 per hour for their participation between 24th January and 28th January 2020. On an initial page before the survey started, participants were informed about the study with a brief summary, the researchers contact information, and information about data protection information. They then could consent with clicking on a buttom before the survey started. At the end of the survey they were further debriefed and they had to click to submit before the end of the study. The LSE Research Ethics Committee approved this study in January 2020. Participants were pre-screened according to their nationality (British), hiring experience and experience of management/supervisory roles. Four participants were excluded as multivariate outliers using Mahalanobis distance ($p < .001$) resulting in a final sample of 183 ($M_{age}$ = 40.96, SD = 9.55; 43.71% women).

Ethnically, 92.89% of our sample identified as White British. Educationally, 45.9% of our sample were educated to degree level, while 24.59% reported a postgraduate degree. All participants reported experience of hiring and on average, reported having evaluated 26.42 CVs or job applications in the past year (SD = 26.16). A between-subject design was used, in which participants were randomly assigned to one of two CV conditions which varied by employment status between employed [105] and unemployed [78]. We conducted a sensitivity analysis using G*Power [24] for a one-tailed t-test with alpha = .05 and power = .80 and can reliably detect effects of $d$ = .373.

### Materials

**Cover story.** Participants were instructed that the purpose of the study was to 'explore evaluations of CVs and what can improve their quality'. They were told that the CV they will see is from a real applicant applying for an assistant manager position and both the job advert and CV are anonymised to protect the anonymity of the applicant and organisation.

**Job advert.** Participants were presented with a real but anonymised job advert for a full-time assistant manager position in a leading fast-moving consumer goods company. The

name of the company is anonymised throughout the advert to avoid confounding the study via associations with 'fit' for a known organisation.

## CVs

One of two CVs were presented to participants randomly and participants were required to view it for at least 45 seconds. The two CVs are identical apart from the dates of employment. In the Unemployed CV, the most recent employment began in March 2016 and ended in December 2017. As such they have ostensibly been unemployed for approximately two years at the time the data was collected. We chose the two year gap because according to data from the Department of Work and Pensions [DWP] in the UK, 67.84% of people claiming Job Seekers Allowance have been claiming for over a year and 26.84% have been claiming for between 2–5 years, more than any other category. As such the two-year duration of unemployment mimics closely the typical scenario for those claiming unemployment benefits.

In the employed condition, the applicants most recent work experience is stated to be March 2016-Present. To equalise the number of years of experience, both candidates have the start date of their first employment varied. In the unemployed condition, the first work experience begins in November 2000–January 2005. In the employed condition, the date is November 2002–January 2005. As such both applicants have an equivalent number of years of experience.

The CVs did not include names and therefore gender, race and other demographic variables can be excluded as possible confounds. The CVs did include the applicant's education, work history and a summary. Of note, the applicants in both conditions are approximately 40 years old (compulsory education completed in 1998). This is in line with data from DWP showing that the typical JSA claimant is between 35–44 years old.

The work experience included in the CV is related to the job on offer and is focused around retail. The organisations the applicants have worked for is anonymised, again to reduce the likelihood that the prestige [or lack thereof] of previous work experience would influence the participants' decision. However, the applicants' experience is not at management level and so the role on offer represents a vertical move in terms of organisational hierarchy. The suitability of the applicant is therefore ambiguous. Nevertheless, both applicants are equivalent, only differing on their current employment status.

## Measures

**Employment-related outcomes.**    Following [25, 26], we asked participants several employment-related questions after they had viewed the CV. All questions were on a 7-point scale from extremely unlikely to extremely likely. Specifically, we asked participants how likely they would be to offer the individual an interview (*Interview*), how likely they would be to want to work with this individual (*Colleague*), how likely they would be to hire the individual (*Hire*), how likely they would be to increase the salary of the individual in the first year (*Salary Increase*) and how likely they would be to promote the individual in the first year (*Promote*). Since the focus of this study is on how likely each candidate might be to get a job, rather than their perceived success in the job, Salary Increase and Promote are not analysed (see Table 1 for descriptive statistics).

**Perceptions of competence, warmth and morality.**    Additionally, we asked participants about the stereotype content they associated with the applicant. These were measured on a 7-point scale from strongly disagree to strongly agree. We asked participants to what extent the applicant seems likeable, friendly, warm (*Sociality*, $a = .84$), trustworthy, sincere, honest (*Morality*, $a = .86$), and intelligent, competent and skilled (*Competence*, $a = .78$). We also

**Table 1. Descriptive statistics for dependent variables as a function of CV.**

| | Employed | | | | Unemployed | | | |
|---|---|---|---|---|---|---|---|---|
| | *M* | *SD* | *Skewness* | *Kurtosis* | *M* | *SD* | *Skewness* | *Kurtosis* |
| Interview | 5.90 | 1.21 | -1.25 | 4.03 | 5.49 | 1.31 | -1.06 | 3.62 |
| Hire | 5.36 | 1.19 | -0.93 | 3.75 | 4.90 | 1.21 | -0.90 | 2.90 |
| Promote | 4.42 | 1.19 | -0.89 | 4.13 | 4.14 | 1.31 | -0.30 | 2.66 |
| Colleague | 5.66 | 0.93 | -0.43 | 2.67 | 5.32 | 0.95 | -1.05 | 3.95 |
| Salary Increase | 4.78 | 1.04 | -0.95 | 5.23 | 4.36 | 1.26 | -0.31 | 2.81 |
| Sociality | 5.17 | 0.80 | -0.04 | 1.95 | 5.07 | 0.70 | -0.02 | 2.17 |
| Competence | 5.74 | 0.74 | -0.37 | 3.18 | 4.51 | 0.63 | -0.66 | 2.98 |
| Morality | 5.46 | 0.81 | -0.16 | 2.24 | 5.33 | 0.77 | -0.04 | 2.26 |
| Salary Offer | £27,435 | £2,190 | 0.83 | 3.11 | £26,700 | £1,551 | 0.72 | 2.52 |

*Note. M* and *SD* represent mean and standard deviation, respectively.

measured the overall stereotype content of the applicant with a 1-item measure 'I have a positive view of the applicant'.

**Salary offer.** We asked participants about the starting salary they would offer the candidate using a sliding scale ranging from £25,000 to £35,000. Participants could select values in £100 increments.

**Attention check.** Finally, we used an attention check to assess whether participants were aware of the applicants' employment status after viewing the CV. Participants were asked 'What is the applicants most recent employment status?'. Those who incorrectly answered this question were deemed to have failed an attention check and were not able to complete the experiment; this includes those who 'timed-out', 'returned' the survey or did not submit a completion code for any reason. Additionally, we asked participants about the perceived education level of the applicant and their perceived age, though these were not used to exclude participants.

## Results

**Employment-related outcomes.** As expected, participants were significantly less likely to want to offer an interview to the unemployed applicant ($M = 5.49$, $SD = 1.31$) compared with the employed applicant ($M = 5.90$, $SD = 1.21$; $t(158.92) = 2.20$, $p = .018$, $d = -.33$ (-.63, -.04)); all results use Welch Correction and Holm-Bonferroni adjusted $p$-values).

Further there was also a significant difference between applicants on participants willingness to hire them. An applicant who was unemployed ($M = 4.90$, $SD = 1.21$) was significantly less likely to be offered a job interview than an applicant who was employed ($M = 5.36$, $SD = 1.19$; $t(164.64) = 2.58$, $p = .016$, $d = -.39$(-.69, -.09)).

Additionally, employment status predicted the likelihood that participants wanted to work with the applicant (colleague); this indicates that an applicant who was unemployed ($M = 5.32$, $SD = 0.95$) was significantly less desirable as a colleague than an applicant who was employed ($M = 5.66$, $SD = 0.93$, $t(164.28) = 2.40$, $p = .018$, $d = -.36$(-.66,-.06).). Employed and Unemployed candidates were offered significantly different salaries as such the unemployed applicant was offered a significantly lower salary than the employed applicant. The means for unemployed ($M = £26,700$, $SD = £1,551$) and employed applicants ($M = £27,435$, $SD = £2,190$) differed by £735; $t(180.63) = 2.66$, $p = .016$, $d = -.38$(-.68,-.08).

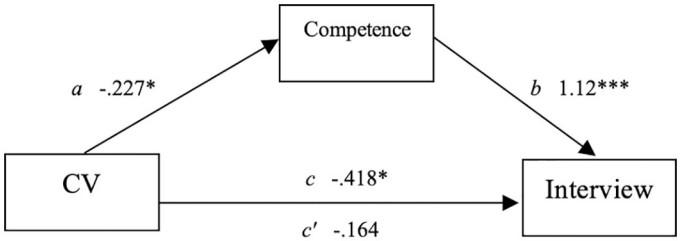

**Fig 1. Unstandarized regression coefficient for the relationship between CV and interviews as mediated by competenence.**

**Mediation model.** Applicants employment status was used to predict the likelihood of being interviewed, with competence expected to mediate the relationship between CV and interview likelihood. See Fig 1 for a visual diagram of the mediated relationship. First, using steps described by Baron and Kenny [27], CV was a significant predictor of interview (the *c* pathway), as shown in Table 2. The unemployed condition showed a lower likelihood of interview than the employed condition, $t(181) = -2.228$, $p = .027$, $\beta = -.418$.

Second, CV was used to predict the mediator, Competence (the *a* pathway) which showed that CV was negatively related to Competence, $t(181) = -2.18$, $p = .031$, $\beta = -.227$. Third, the relationship between the mediator Competence and Interview was examined controlling for the CV (the *b* pathway). Competence was positively related to the likelihood of Interview, $t(180) = 10.6$, $p < .001$, $\beta = 1.12$. Lastly, the mediated relationship between CV and Interview was examined for a drop-in prediction when the mediator was added to the model (the *c'* pathway] Mediation was found, showing that the relationship between CV and Interview was no longer significant after controlling for Competence, $t(180) = -1.10$, $p = .273$, $\beta = -.164$. We tested the significance of this indirect effect using bootstrapping procedures. Unstandardized indirect effects were computed for each of 10,000 bootstrapped samples using the mediation package in R [28]. The bootstrapped unstandardized indirect effect was -.255 [-.496, -.03], $p = .028$ (Fig 1).

**Table 2. Model summaries for mediation analysis.**

| Interview Model | *F* | *p* | $R^2$ |
|---|---|---|---|
| CV predicting Interview | [1, 181] = 4.964 | .027 | .027 |
| CV predicting Competence | [1, 181] = 4.753 | .031 | .026 |
| CV and Competence predicting Interview | [2, 180] = 60.15 | < .001 | .400 |
| **Hire Model** | *F* | *p* | $R^2$ |
| CV predicting Hire | [1, 181] = 6.685 | .010 | .035 |
| CV predicting Competence | [1, 181] = 4.753 | .031 | .026 |
| CV and Competence predicting Hire | [2, 180] = 76.35 | < .001 | .459 |
| **Colleague Model** | *F* | *p* | $R^2$ |
| CV predicting Colleague | [1, 181] = 5.786 | .017 | .031 |
| CV predicting Competence | [1, 181] = 4.753 | .031 | .026 |
| CV and Competence predicting Colleague | [2, 180] = 63.53 | < .001 | .414 |
| **Salary Model** | *F* | *p* | $R^2$ |
| CV predicting Salary | [1, 181] = 6.400 | .012 | .034 |
| CV predicting Competence | [1, 181] = 4.753 | .031 | .026 |
| CV and Competence predicting Salary | [2, 180] = 15.20 | < .001 | .144 |

The same model was tested on the other variables of interest showing equivalent results in each case see Table 2. Thus, for Interview, Hiring, Colleague and Salary Offer the effect of employment status was fully mediated by Competence. In each case the indirect effect was significant using the bootstrapping procedures defined above (Hiring = -.259 (-.505, -.03), $p$ = .027, Colleague = -.192 (-.376, -.01), $p$ = .034, Salary = -214.1 (-453.1, -20.6), $p$ = .032).

Although we expected Competence to be the mediating variable we also tested for differences in morality and sociality between CVs. Two sample t-tests show no differences between the unemployed and employed in terms of either Sociality ($t(176.32)$ = 0.96, $p$ = .294, $d$ = -.14 (-.43, .15)) or Morality ($t(170.62)$ = 1.05, $p$ = .294, $d$ = -.16(-.45, .14)) as such they can both be excluded as possible mediators.

Thus overall, the study supported the three hypotheses. The unemployed candidate was less likely to be interviewed and less likely to be hired than the equivalent employed candidate. This relationship was significantly mediated by perceived competence. In study 2, we provide a preregistered direct replication of these results.

## Study 2

We attempted to replicate the results of study 1 following the same methodology. The study was pre-registered (https://osf.io/krmbq). The hypotheses of study 2 are the same as study 1. We thus predict:

$H_1$—The unemployed candidate will be less likely to be interviewed than an equivalent employed candidate.

$H_2$—The unemployed candidate will be less likely to be offered employment than an equivalent employed candidate.

$H_3$—The relationship between employment status and employment outcomes will be mediated by perceived competence.

Since study 2 is a direct replication of study 1, the methods section only highlights the differences between the two studies.

### Method

**Participants and design.** A priori power analysis was conducted using G*Power [24]. Specifically, we calculated the required sample size of 278 to detect effects of $d$ = 0.3, for a one-tailed t-test [difference between two independent means] with power of .80. As such, 286 participants completed an online experiment on prolific academic (www.prolific.ac) between 12th March and 18th May 2020 and were paid £9.30 per hour for their participation. On an initial page before the survey started, participants were informed about the study with a brief summary, the researchers contact information, and information about data protection information. They then could consent with clicking on a buttom before the survey started. At the end of the survey they were further debriefed and they had to click to submit before the end of the study. The LSE Research Ethics Committee approved this study in January 2020. Participants were pre-screened in the same way as study 1. Eight participants were excluded as multivariate outliers using Mahalanobis distance ($p$ < .001) resulting in a final sample of 278 ($M_{age}$ = 38.35, SD = 9.27; 71.22% women). Readers should note that this data was collected during the hight of the first coronavirus pandemic lockdown in the UK. On May 17th the number of furloughed workers was 8 million compared to 1.3 million in April 2020.

Ethnically, 92.45% of our sample identified as White British. Educationally, 46.4% of our sample were educated to degree level, while 24.82% reported a postgraduate degree. All

**Table 3. Descriptive statistics for dependent variables as a function of CV.**

| | Employed | | | | Unemployed | | | |
|---|---|---|---|---|---|---|---|---|
| | *M* | *SD* | *Skewness* | *Kurtosis* | *M* | *SD* | *Skewness* | *Kurtosis* |
| Interview | 5.79 | 1.22 | -1.40 | 4.85 | 5.48 | 1.37 | -1.03 | 3.25 |
| Hire | 5.30 | 1.24 | -1.00 | 3.58 | 4.97 | 1.33 | -0.67 | 2.49 |
| Colleague | 5.55 | 1.04 | -0.82 | 3.42 | 5.35 | 1.09 | -0.75 | 3.06 |
| Sociality | 5.25 | 0.86 | -0.09 | 2.01 | 4.88 | 0.84 | 0.15 | 2.59 |
| Competence | 5.73 | 0.73 | -0.28 | 3.32 | 5.39 | 0.90 | -0.57 | 3.04 |
| Morality | 5.52 | 0.76 | -0.25 | 2.62 | 5.18 | 0.77 | -0.11 | 2.48 |
| Salary Offer | £27,161 | £2,159 | 0.93 | 3.32 | £27,078 | £2,221 | 1.10 | 3.59 |

*Note. M* and *SD* represent mean and standard deviation, respectively.

participants reported experience of hiring and on average, reported having evaluated 28.43 CVs or job applications in the past year (SD = 26.33). The design of the experiment is the same as the previous study, participants were randomly assigned to either employed [148] or unemployed [130] conditions.

## Materials

**CVs.** The two CVs are identical to those is study one apart from the dates of employment. These are slightly varied to maintain a 2-year gap in unemployment for the unemployed candidate. The employed candidates' dates of employment were equivalently updated.

## Measures

**Employment-related outcomes.** As in study 1, we asked participants several employment-related questions after they had viewed the CV, however, neither Salary Increase nor Promote were not measured in this study (see Table 3 for descriptive statistics).

**Perceptions of competence, warmth and morality.** All stereotype content measures are the same as in study 1 (*Sociality*, a = .89, *Morality*, a = .83, *Competence*, a = .84).

## Results

**Employment-related outcomes.** As expected, participants were significantly less likely to want to offer an interview to the unemployed applicant (*M* = 5.48, *SD* = 1.37) compared with the employed applicant (*M* = 5.79, *SD* = 1.22; $t(260.79) = 1.95$ *p* = .026 (.035), *d* = -.24 (-.47, -.00); Holm-Bonferonni corrected p-value).

Further there was also a significant difference between applicants on participants willingness to hire them, an applicant who was unemployed (*M* = 4.97, *SD* = 1.33) was significantly less likely to be offered a job interview than an applicant who was employed (*M* = 5.30, *SD* = 1.24; $t(265.68) = 2.11$, *p* = .018(.035), *d* = -.269(-.49, -.02); Holm-Bonferonni corrected p-value).

## Mediation model

As in study 1, applicants employment status [employed or unemployed] was used to predict the likelihood of being interviewed, with competence expected to mediate the relationship between CV and interview likelihood. See Fig 2 for a visual diagram of the mediated relationship. CV was a marginally significant predictor of interview (the *c* pathway), as shown in

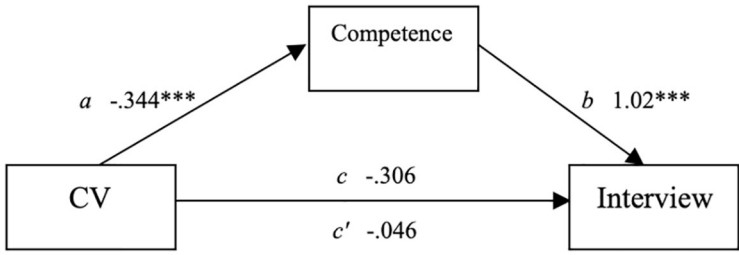

**Fig 2. Unstandarized regression coefficient for the relationship between CV and interviews as mediated by competenence.**

Table 4. The unemployed condition showed a lower likelihood of interview than the employed condition, $t(276) = -1.965$, $p = .050$, $\beta = -.306$.

Second, CV was used to predict the mediator, Competence (the *a* pathway), which showed that CV was negatively related to Competence, $t(276) = -3.495$, $p = < .001$, $\beta = -.344$. Third, the relationship between the mediator Competence and Interview was examined controlling for the CV (the *b* pathway). Competence was positively related to the likelihood of Interview, $t(275) = 14.05$, $p < .001$ $\beta = 1.02$. Lastly, the mediated relationship between CV and Interview was examined for a drop in prediction when the mediator was added to the model (the *c'* pathway). Mediation was found, showing that the relationship between CV and Interview was no longer marginally significant after controlling for Competence, $t(275) = 0.378$, $p = .706$, $\beta = -.046$. We tested the significance of this indirect effect using bootstrapping procedures. Unstandardized indirect effects were computed for each of 10,000 bootstrapped samples using the mediation package in R [28]. The bootstrapped unstandardized indirect effect was -.354 [-.567, -.16], $p = < .001$ (Fig 2).

The same model was tested on Hire showing equivalent results see Table 4. Thus, for Interview and Hiring the effect of employment status was fully mediated by Competence. The indirect effect was significant using the bootstrapping procedures defined above (Hiring = -.365 (-.579, -.16), $p = < .001$].

## Mini-meta analysis of current studies

We conducted a mini meta-analysis of these studies following [29] using fixed effects in which the mean effect size for $H_1$ and $H_2$ was weighted by inverse variance. $Z$ was calculated based on the mean effect size and its standard error. Overall, the difference between employed and unemployed candidates on the interview measure was highly significant $d = .274(.094, .459)$, $Z = 2.912$, $p = .002$, one-tailed. The difference between candidates on the Hire measure was

**Table 4. Model summaries for mediation analysis.**

| Interview Model | F | p | $R^2$ |
|---|---|---|---|
| CV predicting Interview | [1, 276] = 3.861 | .050 | .014 |
| CV predicting Competence | [1, 276] = 12.22 | < .001 | .042 |
| CV and Competence predicting Interview | [2, 275] = 102.1 | < .001 | .426 |
| **Hire Model** | **F** | **p** | **$R^2$** |
| CV predicting Hire | [1, 276] = 4.52 | .034 | .042 |
| CV predicting Competence | [1, 276] = 12.22 | < .001 | .042 |
| CV and Competence predicting Hire | [2, 275] = 118.1 | < .001 | .462 |

also highly significant, $d = .307(.122, .491)$, $Z = 3.253$, $p < .001$. Finally, we performed a meta-analysis of the indirect effect of competence on hiring using the metaSEM package in R [30]. The estimated indirect effect was statistically significant (-.151[-.241, -.062]).

## General discussion

As discussed earlier, unemployed people are a stereotyped group in the UK and elsewhere [31, 32]. They seem to be aware of these stereotypes and report that others see them as less competent than they see themselves [7]. As such, we hypothesized, that perceptions of job candidate's competence would differ as a function of employment status and that the difference in perceived competence would mediate the relationship between employment status and employment-related outcomes.

The present findings support our predictions. We found that perceived competence was predicted by the employment status of the applicant and that perceived competence fully mediated the relationship between the employment status of the applicant and employment-related outcomes. This included the willingness to interview and to hire the candidate. The results were replicated in a high-powered follow-up study which represented a significantly different job market, characterised by increased job insecurity for large parts of society (e.g., through the Covid-19 pandemic). As such all three hypotheses have been supported in two studies and the results appear robust across economic contexts.

As such, we provide evidence that indeed, participants with hiring experience judge unemployed people to be less competent than an employed candidate with equivalent qualifications. Focusing on the role of perceptions of unemployed candidates' competence may help unpack conflicting results in previous audit studies. For instance, [13, 33], show that unemployment status has no effect on employment outcomes for recent graduates. This may be because recent graduates are perceived to occupy a different social identity [34] compared with other unemployed applicants and 'graduates' will likely be seen as relatively competent especially where their most recent experience was as a student compared with unemployed people who are not recent graduates (i.e. whose last experience was not as a student).

Concerning audit studies more generally, we and others [25], have shown that experimental audit-like methods can offer important insights. Data can be obtained that relates to the aims of the audit methodology through online experimental means. Given that the nature and prevalence of bias can change over time, previous audit studies soon fall behind the realities of experienced by different groups. Therefore it can be useful to provide updates about the level of discrimination that different groups face in housing, employment and so on through experimental means.

More broadly, the results of these studies indicate that the mere fact of being unemployed is likely to perpetuate unemployment. This paper provides evidence that knowing the dates of a candidate's employment may lead to bias. The bias against employed candidates is likely to mean the organisation are missing out on talented candidates, whom if employed, would have been shortlisted. Thus, organisations and human resource professionals, in particular, should think differently about the kinds of information that are needed to shortlist applicants. Switching to the length of tenure in each role may alleviate this, whilst still providing the information which is of most use in selecting whom to shortlist, namely the amount of experience they have.

### Limitations and future research

This study and its methodology are not without limitations. It could be argued that the study design does not replicate the typical recruitment scenario where hiring managers and HR

professionals may view dozens of CVs in a short space of time. Under such circumstances (i.e. high cognitive load), research shows that people are more likely to rely on stereotypes [35]. As such, our method might provide a more conservative test of our hypotheses and suggests that the effect of stereotyping unemployed people as less competent is likely to be greater in real-world scenarios.

Moreover, we only use two conditions in this study. Replications with further conditions with differing lengths of unemployment could provide us with an estimate as to the point at which the competence of an unemployed applicant begins to differ significantly from employed candidates.

Furthermore, it seems plausible that perceived competence is not the only factor at play—though perhaps one of the more important ones. Are the differential effects that we see in gender and race audit studies also a matter of competence [36]? Are Black people and women seen as less competent than others? New research will have to be conducted using audit-like experiments to assess these differences.

Finally, new research should explore what practical changes can be made to CVs that would reduce the perception of incompetence. For instance, the audit studies addressing the impact of race have led to names being removed from application forms. Might it be similarly appropriate to remove dates from CVs and only include the duration of any employment alongside a description?

## Conclusion

The current studies provide cause for concern about how stigmatisation affects decision-making in recruitment processes. Across two studies we have shown that unemployed people are less likely to be interviewed and hired compared with an equivalent employed candidate. The reason for this seems to be that unemployed status influences participants perception of the candidates' competence. Knowledge of someone's unemployment alone is not enough to determine whether they are a competent candidate for a job. Yet, the evidence suggests that being unemployed does disadvantage candidates compared to an equivalent employed candidate. If society at large, and employers specifically, want to take advantage of the best available talent then it is important to find ways to reduce bias against unemployed applicants.

## Author Contributions

**Conceptualization:** Celestin Okoroji, Ilka H. Gleibs, Simon Howard.

**Data curation:** Celestin Okoroji.

**Formal analysis:** Celestin Okoroji.

**Investigation:** Ilka H. Gleibs.

**Methodology:** Celestin Okoroji, Simon Howard.

**Supervision:** Ilka H. Gleibs.

**Writing – original draft:** Celestin Okoroji.

**Writing – review & editing:** Celestin Okoroji, Ilka H. Gleibs, Simon Howard.

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
