## [Decision Letter · Decision Letter 0]

24 Oct 2022

PONE-D-22-14505Inferring Incompetence from Employment Status: An Audit-like Experiment

PLOS ONE

Dear Dr. Gleibs,

Thank you for submitting your manuscript to PLOS ONE. After careful consideration, we feel that it has merit but does not fully meet PLOS ONE’s publication criteria as it currently stands. Therefore, we invite you to submit a revised version of the manuscript that addresses the points raised during the review process.

Dear Ilka Gleibs

About your paper entitled “Inferring Incompetence from Employment Status: An Audit-like Experiment” We have considered that the paper is interesting and could potentially be published in a new versión (the decisión is Minor Revision) that takes into account the observations made by the referees.

I am attaching the referee's comments, which will help to explain the reasons for our decision. As you can see, the reviewer finds the paper to be of interest, but raises a number of significant concerns. I hope the reports may be useful if you are considering revising the paper for re-submission to Plos One.

Yours sincerely,

Sonia Brito-Costa

We look forward to receiving your revised manuscript.

Kind regards,

Sónia Brito-Costa, Ph.D.

Academic Editor

PLOS ONE

Journal Requirements:

2. Peer review at PLOS ONE is not double-blinded (https://journals.plos.org/plosone/s/editorial-and-peer-review-process). For this reason, authors should include in the revised manuscript all the information removed for blind review.

3. Please amend your current ethics statement to address the following concerns:

a) Did participants provide their written or verbal informed consent to participate in this study?

5, Please review your reference list to ensure that it is complete and correct. If you have cited papers that have been retracted, please include the rationale for doing so in the manuscript text, or remove these references and replace them with relevant current references. Any changes to the reference list should be mentioned in the rebuttal letter that accompanies your revised manuscript. If you need to cite a retracted article, indicate the article’s retracted status in the References list and also include a citation and full reference for the retraction notice.

Reviewers' comments:

Reviewer #1: I really enjoyed reading this paper. It addresses a timely topic that deserves publication in PLoS ONE, given the methodological rigor based on data availability with R-based codes in a public OSF repository. I only have two concerns regarding the manuscript as it is. First, despite the importance of the topic, the paper omits more recent evidence that should be included in the literature review. In this sense, it is a plus to quote the following studies:

- Garcia-Lorenzo, L., Sell-Trujillo, L., & Donnelly, P. (2021). Responding to stigmatization: How to resist and overcome the stigma of unemployment. Organization Studies, 01708406211053217.

- Suttill, B. (2021). Non-academic, lazy and not employable: Exploring stereotypes of NEETs in England. Employment and Careers.

The second concern relates to the communication strategy followed in the article. For example, although Table 1 and 3 summarize descriptive statistics of each study, they omit third and fourth statistical moments (asymmetry and kurtosis) that might be informative for a general audience if communicated with ridgeplots. Given the fact that authors analyzed their data with R-codes, it would be nice to see statistical differences visually.

6. PLOS authors have the option to publish the peer review history of their article (what does this mean?). If published, this will include your full peer review and any attached files.

Reviewer #1: **Yes: **Juan C. Correa (ORCID:0000-0002-0301-5641)

---

## [Author Response · Author response to Decision Letter 0]

21 Nov 2022

Dear Dr Brito-Costa, 

Thank you very much for your positive response in the first round of revision. We were very encouraged by your decision letter and the reviewer’s comments. We explain below how we dealt with the individual points mentioned. 

We have now updated the document and use the PLOS One style template throughout. 

2. Peer review at PLOS ONE is not double-blinded (https://journals.plos.org/plosone/s/editorial-and-peer-review-process). For this reason, authors should include in the revised manuscript all the information removed for blind review.

We have now included the title page with all authors identified and unblinded the references.

3. Please amend your current ethics statement to address the following concerns:

a) Did participants provide their written or verbal informed consent to participate in this study?

We used an online survey where people read the informed consent before they had to agree/disagree with the survey before they could continue. We have also included information that the LSE ethics committee approved the study. 

We will be able to provide a DOI number for the data, which is Identifier: DOI 10.17605/OSF.IO/RF7HW

We will be able to update the Data Availability statement with this. 

 Garcia-Lorenzo, L., Sell-Trujillo, L., & Donnelly, P. (2021). Responding to stigmatization: How to resist and overcome the stigma of unemployment. Organization Studies, 01708406211053217.

- Suttill, B. (2021). Non-academic, lazy and not employable: Exploring stereotypes of NEETs in England. Employment and Careers.

Thank you very much, we have included these in the manuscript; we are not citing any work that was retracted. 

6. The second concern relates to the communication strategy followed in the article. For example, although Table 1 and 3 summarize descriptive statistics of each study, they omit third and fourth statistical moments (asymmetry and kurtosis) that might be informative for a general audience if communicated with ridgeplots. Given the fact that authors analyzed their data with R-codes, it would be nice to see statistical differences visually.

Thank you also for this comment; we have now included the information on symmetry and kurtosis in the descriptive tables. We have not included a ridgeplot as we felt is was not necessary to include more Figures. 

In sum, given our revision, we hope that you find the revised manuscript to be of wide interest and suitable for publication in PLoS One and we look forward to your editorial response.

Sincerely, 

Dr Ilka Gleibs as corresponding author on behalf of all authors.

---

## [Editor Report · Decision Letter 1]

4 Jan 2023

Inferring Incompetence from Employment Status: An Audit-like Experiment

PONE-D-22-14505R1

Dear Dr. Ilka H. Gleibs,

We’re pleased to inform you that your manuscript has been judged scientifically suitable for publication and will be formally accepted for publication once it meets all outstanding technical requirements.

Kind regards,

Sónia Brito-Costa, Ph.D.

Academic Editor

PLOS ONE

---

## [Editor Report · Acceptance letter]

6 Jan 2023

PONE-D-22-14505R1 

Inferring Incompetence from Employment Status: An Audit-like Experiment 

Dear Dr. Gleibs:

I'm pleased to inform you that your manuscript has been deemed suitable for publication in PLOS ONE. Congratulations! Your manuscript is now with our production department. 

Kind regards, 

on behalf of

Dr. Sónia Brito-Costa 

Academic Editor

PLOS ONE